# A One Health Approach for Guinea Worm Disease Control: Scope and Opportunities

**DOI:** 10.3390/tropicalmed5040159

**Published:** 2020-10-13

**Authors:** Matthew R. Boyce, Ellen P. Carlin, Jordan Schermerhorn, Claire J. Standley

**Affiliations:** Center for Global Health Science & Security, Georgetown University, Washington, DC 20057, USA; ec1223@georgetown.edu (E.P.C.); js4904@georgetown.edu (J.S.); Claire.Standley@georgetown.edu (C.J.S.)

**Keywords:** animal health, *Dracunculus medinensis*, guinea worm, human health, infectious disease, one health, zoonoses

## Abstract

Guinea worm disease (GWD) is a neglected tropical disease that was targeted for eradication several decades ago because of its limited geographical distribution, predictable seasonality, straightforward diagnosis, and exclusive infection of humans. However, a growing body of evidence challenges this last attribute and suggests that GWD can affect both humans and animal populations. The One Health approach emphasizes the relatedness of human, animal, and environmental health. We reviewed epidemiological evidence that could support the utility of a One Health approach for GWD control in the six countries that have reported human GWD cases since 2015—Angola, Cameroon, Chad, Ethiopia, Mali, and South Sudan. Human GWD cases have dramatically declined, but recent years have seen a gradual increase in human case counts, cases in new geographies, and a rapidly growing number of animal infections. Taken together, these suggest a need for an adjusted approach for eradicating GWD using a framework rooted in One Health, dedicated to improving disease surveillance and in animals; pinpointing the dominant routes of infection in animals; elucidating the disease burden in animals; determining transmission risk factors among animals and from animals to humans; and identifying practical ways to foster horizontal and multidisciplinary approaches.

## 1. Introduction

Guinea worm disease (GWD), also known as dracunculiasis or Medina worm disease, is a neglected tropical disease (NTD) caused by a nematode, *Dracunculus medinensis*. Humans usually become infected by ingesting water containing copepods (small aquatic crustaceans) infected with Dracunculus larvae [1]. While the copepods are killed in the stomach, the larvae mature into adult Guinea worms and copulate. Male worms die following copulation, but approximately one-year post-infection, adult female Guinea worms migrate to the skin and breach the surface through a blister. When these wounds are exposed to a water source, the female worm releases embryos (i.e., L1 larvae) and the cycle repeats. 

Secondary infections are common, and complications can result in abscess formation, tetanus, septic arthritis, or systemic sepsis [1]. Though rarely fatal, GWD can cause permanent disability and impose major economic burdens [1,2]. Like many other NTDs, GWD primarily affects poor, rural communities. Neither a vaccine nor effective medical prophylaxis is currently available [3,4]. Case management is limited to carefully and safely removing the whole worm and tending the exit wound to prevent infection.

Following the success of the smallpox eradication campaign in 1979, public health experts identified GWD as a candidate for eradication [4]. GWD was considered a suitable candidate because its geographical distribution was limited to tropical or subtropical areas, it adhered to seasonal transmission patterns [5,6], the diagnosis was straightforward [7], and, critically, there were no known animal reservoirs [3,4]. 

Led by the Carter Center and supported by the World Health Organization, United Nations Children’s Fund, and United States Centers for Disease Control and Prevention (CDC), the Global Dracunculiasis Eradication Campaign began in 1980 and has significantly reduced the global burden of the disease [8]. Since it began, when there were more than 3 million cases of GWD in 20 countries, the eradication campaign has dramatically reduced GWD prevalence and as few as 22 cases were reported in four countries in 2015 [9]. However, recent years have seen an increase in the number of reported cases, as well as human cases and animal infections reported in new countries and countries that had previously been certified free of GWD. Consequently, the eradication target has been delayed until 2030 [10].

These delays are the result of numerous factors, including conflict and social unrest, that have limited the ability of field teams to conduct active searches for cases [2,10,11,12]. However, an emerging body of evidence also suggests that eradication efforts have been further impeded by the presence of other non-human animal species (hereafter “animals”) that act as hosts and maintain transmission. Dogs, cats, and baboons, and potentially fish and frogs, are now recognized as playing a role in the parasite’s transmission to human populations [3,13,14,15,16,17,18,19]. Of these, widespread reservoir infections in dogs are considered the largest threat to eradication efforts [11,20]. Adult Guinea worms emerging from humans and these other animals are genetically indistinguishable, with evidence that worms emerging from animals have subsequently caused infections in humans by releasing larvae into shared water sources [13].

That multiple countries now report animal infections underscores the need for novel intervention and eradication strategies [21]. The identification of the worm in so many species complicates eradication and necessitates an interdisciplinary approach. One Health is a concept that acknowledges the relatedness of human, animal, and environmental health and the importance of interdisciplinary efforts in achieving positive health outcomes across these populations [22,23]. Although the body of work to eradicate GWD has been substantial, relatively little research has been undertaken to elucidate the role that animals may play in transmission or to isolate the impact of specific interventions on the number of animal infections. Accordingly, this paper reviews the epidemiological evidence as it relates to animal epidemiology, and in turn to potential animal-level interventions toward the goal of human eradiation, and creates a conceptual framework for how such an approach could be applied to create new GWD control paradigms that support eradication efforts.

## 2. Materials and Methods

We conducted a review to identify GWD surveillance data for humans and animals and to identify interventions being implemented to control the spread of disease. We performed a literature review using the PubMed and Web of Science databases. Synonyms for “Dracunculiasis”, “dog”, “cat”, “baboon”, “frog”, and “fish” were combined with a list of the countries that have reported GWD cases since 2015 (i.e., Angola, Cameroon, Chad, Ethiopia, Mali, and South Sudan). A Appendix A provides the complete search syntax (Appendix A). Our last search was conducted in January 2020.

We attempted to obtain detailed monthly human and animal surveillance data, including through outreach to the Carter Center, but found no sources. To address this, we collected all Guinea Worm Wrap-Up reports—reports periodically co-published by the Carter Center and the U.S. CDC that contain human and animal surveillance data—published between January 2015 and January 2020 and reviewed their surveillance data for human cases and animal infections [24]. The monthly human surveillance data were robust, but the monthly animal surveillance data were not publicly reported. As a result, for animals, we gathered monthly surveillance data when available, but focused efforts primarily on annual data. These were supplemented with data from the World Health Organization. Gathered data were compiled in a Microsoft Excel spreadsheet (Appendix A).

We then reviewed and analyzed the identified peer-reviewed literature and surveillance data to synthesize a conceptual framework for how such an approach that emphasizes the nexus of human, animal, and environmental health could enhance efforts to eradicate GWD.

## 3. Results

### 3.1. Epidemiological Data

In 2010, a collective 1786 human GWD cases were reported in the five countries considered in this study, with a majority of these coming from South Sudan (Table 1). By 2015, total reported human cases declined to 22, from which point they gradually rose to reach 53 in 2019 (Figure 1), with a majority of these cases coming from Chad. Four countries reported human GWD cases from 2010–2015, three from 2016–2018, and four in 2019. Angola and Chad were the only two countries to report more human GWD cases in 2019 than in 2010. After no previous reports of cases (with the caveat that no GWD-specific case search programs had ever been active in the country), Angola reported one human GWD case in 2018 and another in 2019 (Table 1). Chad reported 10 human cases in 2010, and a low of 9 cases in 2015, before reported cases began to increase, reaching 47 in 2019.

A variety of mammalian species have long been established as potential reservoirs of GWD; however, there was limited evidence that these species were transmitting Guinea worm to humans [3,15]. Sporadic infections in dogs and other animals have been historically reported in Cote d’Ivoire, Ghana, India, Kazakhstan, and Pakistan, but studies to definitively assess animals as a transmission risk factor for humans are limited. In the countries listed, infections in animals are thought to have ceased after human infections were eliminated [28]. Whether these animal infections were instances of anthroponosis (i.e., the transmission of a pathogen from humans to a nonhuman species) that were then interrupted via interventions in the human host is not known as studies have not assessed this question.

There has been an increase in reported animal Guinea worm infections since 2015. Most reported animal Guinea worm infections have involved dogs in Chad. The total number of reported canine infections has grown from 498 in 2015 to 1943 infections in 2019 (Table 2). The pathways by which dogs are infected remain unconfirmed, but it is hypothesized they can acquire Guinea worm infection in the same way as humans (i.e., from drinking water containing infected copepods), or possibly by consuming infected paratenic hosts [15]. The consumption of raw fish entrails, discarded by humans during food preparation, has been considered a potential source of transmission and addressed by the human behavioral intervention of promoting burial of entrails in all known-endemic villages since 2015 [21]. A study conducted in Chad found that dogs in households where water was provided for animals were less likely to have had Guinea worm and that dogs with a higher proportion of fish in their diets were more likely to have had Guinea worm [19]. However, dogs with less fish in their diets and those more likely to consume provided water may also live further from natural water sources and therefore less likely to be infected via the traditional waterborne route. A similar study conducted in Ethiopia found no evidence to support the hypothesized paratenic host transmission pathway [29]. Further, although the phenomenon of canine infections with Guinea worm was only recently realized, evidence suggests that this was the result of insufficient surveillance rather than the absence of infection [13].

Guinea worm infections have also been reported in cats, baboons, a leopard [27], and a donkey [28]. The incident involving the leopard was later removed from official counts because the worm was found while dissecting the dead animal, and had not emerged as is required for official case counts [27,30]. Excluding dogs, animal infection counts rose from 5 reported infections in 2015 to 53 in 2019 (Table 2).

Chad has reported the most animal infections, including dogs, of any country and there appears to be a temporal alignment between patterns of reported human cases and animal infections (Figure 2). After first reporting 27 infections in dogs in 2012 [32], the number has risen steadily, peaking at 1927 infections in 2019 [30]. Humans and dogs are infected by shared populations of worms in Chad and it has been hypothesized that this may be driving the increase in human GWD cases [13,15]. Chad also reported four Guinea worm infections in cats in 2015, rising to 46 infections in 2019 following intensified education and surveillance efforts targeting cats.

Ethiopia reported one Guinea worm infection in baboons annually from 2013–2015 [28,33], two infections in 2016 [34], four infections in 2017 [35], one infection in 2018 [36], and six infections in 2019 [30]; Ethiopia also reported five infections in cats in 2018 [36]. Mali reported one infection in a cat in 2017 [35], two infections in 2018 [31], and one infection in 2019 [30]. From 2015–2019, Angola and South Sudan each reported one Guinea worm infection in dogs [30,35], and neither has reported infections in other animals. 

### 3.2. Interventions Used to Reduce the Burden of Guinea Worm Disease in Humans and Animal Populations

GWD eradication efforts primarily rely on behavioral- and environmental-based interventions and associated health education campaigns to break the cycle of transmission between humans and the environment [37]. Countries that are endemic for GWD often endorse layered approaches that emphasize the implementation of multiple interventions with the understanding that no single intervention will be perfectly executed [38].

Many GWD eradication programs are based on the provision of improved water sources for human populations [4,18,20,39,40] (Figure 3). Specific interventions include the use of water filtration devices [4,18,20,39,40] and providing clean water from underground sources by hand-dug or bore-hole wells or other similar means [4,18,39,40]. Sand, steel mesh, and finely woven cloth water filtration devices prevent GWD in humans by removing the copepods from drinking water. Portable pipe filters—straws with a filter inside—allow people to filter their water when traveling or away from home [40]. A more communal solution involves drilling boreholes to provide clean water from underground sources, which protects against other infectious diseases in addition to GWD [40].

To supplement interventions focused on drinking water, eradication campaigns also treat water sources with larvicides to reduce and control vector populations. Temephos is a colorless, odorless, and tasteless larvicide that is harmless to humans, fish, and plants at recommended concentrations, but lethal to the copepods that act as intermediate hosts for Guinea worm [40]. Accordingly, it is frequently used to treat contaminated or potentially contaminated sources of surface water to reduce transmission risk [2,3,4,12,18,20,40,41,42]. It may be proactively applied at monthly intervals [4], or reactively following the known contamination of a water source by an infected human or animal within 10–14 days [28]. 

Challenges related to treating water sources with temephos include the intensity of labor, difficulties in reaching remote areas, difficulties in identifying which surface water sources require treatment, and effectively treating larger or flowing bodies of water (which is generally less successful than treatments of smaller or stagnant bodies) [3,4,18]. Because animals may infect and utilize water sources that are not used by humans, some programs endeavor to map and apply larvicide to hidden water sources in at-risk areas; however, seasonal flooding and the drying of many water sources make this difficult to track [2,19]. Overall, there is some evidence that suggests vector control interventions are an effective method for suppressing GWD infections via the traditional water-based infection route; in theory, it should also reduce hypothesized paratenic transmission if the infected copepods are killed prior to consumption by paratenic hosts.

Other interventions focus on the containment of human and animal cases before they have an opportunity to enter and contaminate water sources [14,18,20,41,42]. Eradication programs have promoted the tethering of domestic dogs and cats with signs of infection, to prevent them from contaminating water sources once the worms fully emerge; Ethiopia and Chad have piloted preventative tethering of animals without symptoms in villages identified as high-risk [14,41,42]. Eradication programs have provided monetary incentives for complying with tethering protocols; the Ethiopia program has also imposed fines for persons found releasing animals from tethering prematurely [43]. GPS collars have also been piloted to track the movement of baboons [19,36,44], with the goal of monitoring the potential contamination of water sources even in densely forested areas [45].

To increase the sensitivity of surveillance efforts, programs have also introduced cash rewards for reporting suspected cases of GWD in animals and humans that are subsequently confirmed [2,4,18]. This form of active surveillance depends on the rapid investigation of rumors, a robust network of staff and volunteers, and a reliable reporting system [20]. Rewards are generally higher for reporting human cases of GWD and generally increase as countries report fewer human cases. Cash rewards for reporting infections in dogs are occasionally split between the person who reports the dog and the owner of the dog, in exchange for tethering the animal until the worm emerges [35,46,47,48].

Health education and awareness campaigns are also used to reduce transmission. Campaigns frequently focus on educating people about the transmission cycle, how the infection is acquired, and how it can be prevented [2,18,20,21,40,41]. These efforts are designed to inform people about the aforementioned interventions but have also encouraged other behavioral interventions, such as sufficiently cooking, drying or smoking hypothesized paratenic hosts before eating [2,18,21,41], and burning or burying fish entrails that may contain infected copepods to prevent animals from eating them [2,18,21,41,42]. Campaigns occur in person, through radio and television programming, via printed materials (e.g., posters), and through existing community structures like school systems and houses of religion to achieve maximum reach [48]. These efforts are of importance in endemic communities, neighboring communities, and at-risk non-endemic areas [2].

Some additional interventions have aimed at using medical treatment to prevent or cure infections in dogs. Topical and oral anthelmintics commonly used in dogs as preventives for other nematode infections, such as heartworm, have been tested with limited effects [38,48]. Trials of ivermectin and imidacloprid/moxidectin have not demonstrated effectiveness in preventing or curing infections in dogs and were discontinued [49]; an oral trial with flubendazole is ongoing [38,50]. These recent, limited efforts at incorporating veterinary treatment into human eradication goals may benefit from more laboratory research prior to field trials and, if successful in controlled environments, incorporation into a systemic, long-term interdisciplinary strategy. 

## 4. Discussion

Pathogens with complex transmission cycles that involve multiple hosts can be exceptionally challenging to eliminate and eradicate. The massive mid-21st century campaign to eradicate smallpox was successful because of the comparatively simple, single-host epidemiology of the smallpox virus. Most pathogens of humans are characterized by more complex transmission pathways. Indeed, most infectious diseases of humans have zoonotic origins [51], meaning that, by definition, animals can serve as reservoirs, paratenic or dead-end hosts, or transmitters of the pathogen in question.

GWD, once believed to have been exclusively a disease of humans, has proven capable of infecting other animals. As the lack of zoonotic reservoirs has traditionally been considered a key factor for identifying human diseases with the potential for eradication [52]; this begs the question of whether GWD should no longer be considered eradicable; whether GWD should be considered eradicable, but only with respect to the disease in humans, and not entirely removed from the ecosystem (i.e., alter the current definition of “eradicated”); or pursue the full eradication of Guinea worm across both humans and animals, and use it as evidence that the previous tenets of “eradicable” are flawed. We advocate for the last of these options and believe that more intensive and holistic interventions and approaches that consider environmental, human, and animal elements as part of an interacting whole are more likely to be successful. 

The veterinary profession has targeted prevention and treatment of gastrointestinal worms in dogs and cats as a way not only to keep pets healthy but also to prevent transmission of worms to humans. There is some evidence for success. For instance, higher-income nations often report much lower rates of toxocariasis than lower-income nations [53]. Yet, *Toxocara* seroprevalence and other indicators remain alarmingly high, even in the industrialized world, in part because of earlier misconceptions about the parasite’s lifecycle, lack of access to animal care, and practical challenges of treating street dog populations—all challenges similar to those associated with GWD. 

Rabies represents another example of successful coordination between human and veterinary health specialists working in lock-step toward the common goal of eliminating rabies in human populations. Many countries have had success with a case management mechanism that spans human and animal healthcare-seeking known as integrated bite case management [54,55]. In addition, countries that have eliminated rabies in people often have dedicated and aggressive canine and feline rabies vaccination programs implemented through legal requirements on pet owners to vaccinate their animals. Indeed, the World Health Organization’s strategy for eliminating human rabies in countries where it remains endemic is to eliminate it in dogs through mass vaccination campaigns [56].

While existing interventions for GWD have generally been successful in controlling the disease in human populations, they have been unsuccessful in halting transmission, as infections in animals have recently been detected in all of the remaining endemic countries. The increasing presence of animal infections suggests that improved adherence to existing interventions and/or new interventions targeting animals are necessary [2], especially given that people will likely be at risk so long as the parasite continues to infect animals. Zoonotic diseases, like GWD, may therefore benefit from strategies that explicitly integrate public health, veterinary medicine, animal management, and ecological approaches [57].

A renewed approach to eradicating GWD in affected countries can be achieved through a framework and set of targeted activities that improve disease surveillance in humans and animals; pinpoint the dominant routes of infection in animal hosts; elucidate the burden of disease in animals; determine risk factors for transmission among animals and from animals to humans; and identify practical ways to foster horizontal and multidisciplinary intervention approaches.

The eradication of GWD will require a holistic understanding of the burden of disease and the identification and containment of every case. While daunting, as mentioned earlier, this is necessary because humans and animals are infected by shared populations of Guinea worms, and as a result, humans be at risk for infections so long as the parasite continues to infect animals. This makes disease surveillance, case finding, and case containment a priority [58]. In Chad, a consistent pattern has been the discovery of dog infections with the expansion of active surveillance following identified human cases in new areas. For instance, after 13 human cases were detected in the Moyen-Chari region of Chad between 2010 and 2014, the initiation of active surveillance in 2015 uncovered hundreds of infected dogs in subsequent years; the region reported 409 dog infections in 2018 and 855 dog infections in 2019 [59,60]. A similar discovery may come to pass in Chad’s Tandjile region, which only initiated active surveillance in 2019 following sporadic detection of human cases since 2011 [38]. Surveying communities about the year they first noticed worms emerging from dogs—and comparing responses to initiation of active surveillance programs—may provide further insights. Given the relative numbers, it seems more likely that transmission among dogs is driving cases in humans rather than vice-versa, and active surveillance with an equal focus on animal transmission should immediately follow the detection of human cases in new areas. Because dogs, unlike humans, are not able to report their own infections, exploratory surveillance efforts outside of areas known to be endemic should also be considered. These efforts should include periodic capture and examination of free-ranging and feral dogs in areas known to be endemic, as these animals may perpetuate transmission even when infections in domestic dogs are fully contained.

To this end, identifying practical ways to foster horizontal and multidisciplinary approaches could improve surveillance and control efforts. Coordinating interventions with other human and animal public health initiatives, and leveraging existing One Health efforts, represent opportunities to achieve this goal. For example, in Mali, NTD control efforts use predominantly vertical approaches, and while there is a One Health platform, it is not currently involved in the Guinea worm effort. Integrating surveillance activities in humans and animal populations could ensure more routine activities for both populations also help to ensure that GWD is not reintroduced to areas where it had previously been eliminated. Further, integrating NTD control programs with other health programs has proven to be a practical option for improving accountability, efficiency, and cost-effectiveness [61], and both of the human cases in Angola and Cameroon were initially recognized by healthcare workers who were involved in national immunization campaigns [27,35]. Future work should investigate the cost-effectiveness of current strategies, as well as that of new interventions to help determine which approaches for eradication are most appropriate for areas that remain at risk.

With regard to surveillance in animal populations, integrating Guinea worm and rabies could represent a feasible option, as some countries have robust rabies vaccination programs that could also be used to conduct Guinea worm infection exams. Integrating GWD control efforts with these initiatives may improve the reach, cost-effectiveness, and sustainability of surveillance for dog infections in particular, and may help flag areas endemic for animal transmission for expansion of both active surveillance and control interventions before human cases appear. Different levels of integration could include surveying rabies vaccination teams for reports of rumored dog infections; training rabies teams to identify symptoms, report infections, and inform dog owners about reward money; or, most intensively, providing rabies field staff with sample collection tubes and reward money for identifying confirmed infections in areas beyond the reach of active surveillance. Further, for countries that do not have robust rabies control programs, integrating GWD and rabies control efforts could incentivize collaboration between veterinary and human health departments and correspondingly improve rabies vaccination rates—which would benefit both GWD and rabies control efforts.

Beyond simply detecting infections, understanding relevant risk factors for transmission and how transmission is occurring among dogs, cats, and other animal species would allow for the implementation of timely and more effective control measures. It is possible that animals, and in particular dogs, are exposed to Guinea worm infection by both foodborne and waterborne routes of transmission, or through some combination that amplifies infection in non-human hosts [19]. Although recent research efforts have focused on possible infection of dogs via consumption of raw or undercooked paratenic hosts infected with the parasite [15,20], targeted searches revealed a very low prevalence of infection among frogs and no Guinea worms detected in dissected fish [62]. This may point to waterborne transmission as the dominant route of infection in dogs [17]. This is especially true when comparing against the high infection rates among dogs in the villages in which the fish and frogs were obtained and the frequency of frog consumption among dogs. This hypothesis may also be supported by the observation of year-over-year clusters of worm emergence during short periods of time, due to staggered ingestion of infected paratenic hosts. Examining dog infection line lists may provide further clues and the continued evaluation of transmission routes could help refine or devise interventions [42]. Different approaches may also be required for different animal populations (e.g., domestic versus free-ranging dogs) and it will be important to identify risk factors to inform control efforts [19]. Community surveys about the presence of free-ranging or feral dogs, or wild dogs and cats, in endemic areas could also inform approaches and help to elucidate whether or to what extent these populations play a role in sustaining transmission.

Eradication of any infectious disease requires simple diagnostics, simple preventives and/or treatments, and political will. The growing scientific understanding of the zoonotic capacity of Guinea worm underscores the importance of diagnosis in non-human hosts. Diagnosis is key to enabling proper epidemiological understanding and to inform prevention and treatment decisions. At present, diagnosis in dogs is done in the same manner as for people (i.e., visual detection of the worm at the skin surface). This diagnostic approach allows for detection only at a very late stage of infection. A field-based rapid diagnostic assay such as antibody serology to assess pre-patent infection in dogs could support improved understanding of canine prevalence, and intervention decisions like proactive tethering. Hand-held ultrasound machines, frequently used in equine medicine, could also be used to detect the worms in advance of emergence. Identified infected dogs could then be prioritized by monitoring programs through routine checks for worm emergence. Some of the worms identified on ultrasound could potentially be removed surgically by qualified personnel, such as spay-neuter teams, where such resources are available. Improved diagnostics when combined with an interdisciplinary research and surveillance approach could shift the research paradigm toward a more comprehensive understanding of infection prevalence in other animals. For a less technical approach, routine spot-checks of dogs for signs of worm emergence, either conducted by program staff or owners, may increase early detection and containment of Guinea worms.

Similarly, new tools for environmental surveillance could be useful for targeting limited health education and surveillance resources toward communities and at highest risk. An assay developed to amplify Guinea worm DNA successfully identified larvae from infected copepods present in samples of pond water taken from Chad [63]. While such a tool could potentially shed light on the temporal and spatial distribution of larvae in the environment, a field-informed sampling strategy and a close collaboration with community health workers and water treatment teams would be needed to operationalize the results.

Community responses, participation in, and behavior change stemming from these activities should also be monitored [64], as such considerations will impact overall intervention effectiveness and could allow for eradication programs to tailor intervention strategies to specific sub-populations. For instance, monthly spot-checks of fish entrail burial sites have been conducted in villages under active surveillance in Chad since 2015, with consistent reports of high adherence [27,43,46]. If fish entrail burial is not associated with reductions in animal infections, further analysis of related human and animal behaviors may uncover alternate, more effective interventions, or suggest a different primary route of infection. 

Beyond One Health, other aspects that require multisectoral approaches also challenge eradication efforts. For instance, conflict and migration present challenges for containing any infectious disease, but have notably hindered surveillance and control efforts for Guinea worm in both human and animal populations [12]. A new human case in Mali in 2020, the first in four years, may indicate conflict-related surveillance gaps [65]. In the Malian context, insecurity has also limited program operations in regions of the country where dogs are bred and become infected before being transported to other areas [36]. The Guinea Worm Eradication Program of South Sudan is taking action to address some of these challenges by tracking the migratory patterns of individuals associated with cattle camps, which characterized the Guinea worm cases in 2018 [59]. Similar efforts may prove beneficial in Chad, where dogs and humans seasonally migrate to fishing grounds, and where tracking these patterns may help identify new areas of endemicity.

## 5. Conclusions

The decades-long effort to eradicate GWD has been delayed as a result of contextual challenges but also because infections in non-human animal species are complicating control efforts. While existing strategies and interventions have achieved impressive decreases in human GWD cases, an apparent epidemiological shift occurred over the past decade whereby the largest burden of GWD shifted from human populations to animals. Although this shift likely reflects changes in surveillance and not novel transmission patterns in other species, it underscores the importance of embracing new approaches if the goal of eradicating GWD is to be realized. Accordingly, a One Health approach that seeks to improve surveillance efforts in human and animal populations, elucidate the burden of disease in animal populations, identify risk factors for transmission among human and animal populations, and foster horizontal and multidisciplinary interventions could represent one way to reach the elusive but attainable goal of eradicating GWD in humans and animals.

## Figures and Tables

**Figure 1 tropicalmed-05-00159-f001:**
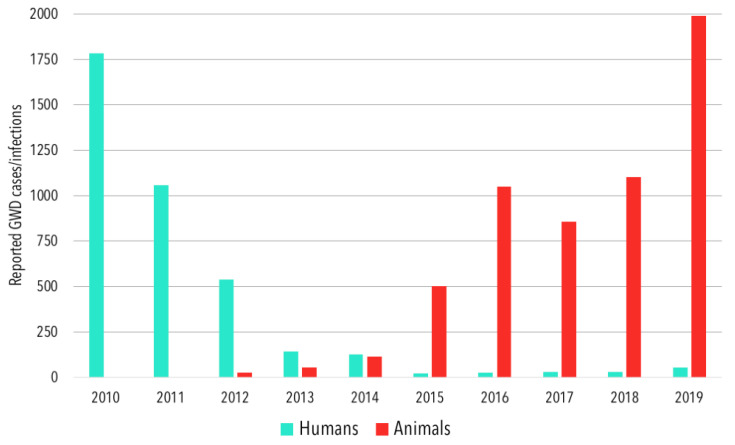
The total annual reported human cases and animal infections of Guinea worm disease in Angola, Cameroon, Chad, Ethiopia, Mali, and South Sudan, 2010–2019.

**Figure 2 tropicalmed-05-00159-f002:**
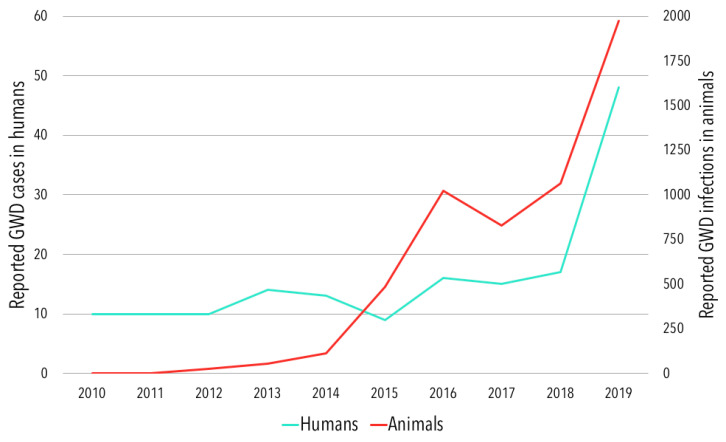
Reported human cases and animal infections of Guinea worm disease in Chad, 2010–2019.

**Figure 3 tropicalmed-05-00159-f003:**
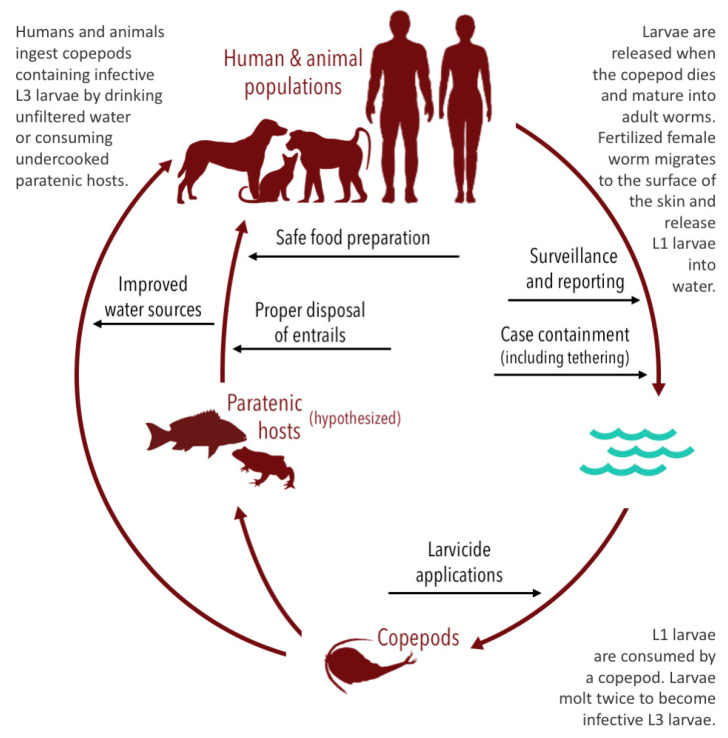
Transmission and life cycle of the Guinea worm with impact points of existing interventions.

**Table 1 tropicalmed-05-00159-t001:** Reported Human Cases of Guinea Worm Disease in Considered Countries, 2010–2019.

Year	Country						Total
	Angola	Cameroon	Chad	Ethiopia	Mali	South Sudan	
2010	0	0	10	21	57	1698	1786
2011	0	0	10	8	12	1028	1058
2012	0	0	10	4	4	521	539 ^1^
2013	0	0	14	7	11	113	145 ^2^
2014	0	0	13	3	40	70	126
2015	0	0	9	3	5	5	22
2016	0	0	16	3	0	6	25
2017	0	0	15	15	0	0	30
2018	1	0	17	0	0	10	28
2019	1	1 ^3^	47	0	0	4	53

^1^ An additional three cases were reported in Niger in 2012, but none have been reported since. These were thought to have been imported from Mali [25]. ^2^ An additional three cases were reported in Sudan in 2013, but none have been reported since. While the cases were not classified as imported, they were detected in an area of southwest Sudan that borders South Sudan and the Central African Republic [26]. ^3^ A 2019 case in Cameroon was found close to the Chad border; the origin of infection remains unclear, though human cases and reports of animal infections from the same part of Cameroon could suggest localized transmission [27].

**Table 2 tropicalmed-05-00159-t002:** Reported Animal Infections of Guinea Worm Disease in Considered Countries, 2015–2019; (Canine Infections).

Year	Country						Total
	Angola	Cameroon	Chad	Ethiopia	Mali	South Sudan	
2015	0	0	487 (483)	14 (13)	1 (1)	1 (1)	503 (498)
2016	0	0	1022 (1011)	16 (14)	11 (11)	0	1049 (1036)
2017	0	0	830 (817)	15 (11)	10 (9)	0	855 (837)
2018	0	1 (1)	1065 (1040)	17 (11)	20 (18)	0	1103 (1070)
2019	1 (1)	5 (5) ^1^	1973 (1927)	8 (2)	9 (8)	0	1996 (1943)

^1^ This number includes only reported infections explicitly mentioned in a Carter Center update [31], and does not include other unconfirmed rumors discovered by WHO investigators.

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
