# Peer review of "A One Health Approach for Guinea Worm Disease Control: Scope and Opportunities"

_tropicalmed, 2020, doi:10.3390/tropicalmed5040159_

Round 1

Reviewer 1 Report

Line 30: the larval stage corresponds to the parasite (Dracunculus) not to the disease (dracunculiasis).

Line 33: to avoid confusion with the larvae inside the copepod, or in line 30 "L3 larvae" and in line 33 "L1 larvae” is specified, or “larvae” in line 33 should be replaced by "embryos".

Line 38: The highlighted sentence should be changed. Certainly, there is no vaccine or effective treatment, but, of course, there have been and there are several prophylactic measures to prevent GW disease. As a matter of fact, thanks to those effective prophylactic measures adopted by the GW eradication campaign (explained by the authors in 3.2. section), GWD was eradicated in the majority of the countries affected by the disease in the 1960's.

Line 167: When listing the interventions implemented to reduce the burden of GWD in section 3.2., it would be more comprehensible if it were made in chronological order, i.e., with a clear distinction between the classical measures (those related to obtaining safe water) and those implemented very recently, due to, in turn, the recently accepted role of animals as reservoirs of the disease (those from Line 202: Tethering dogs and cats, the rewards, etc).

Line 337: this assertion was already made by Galán-Puchades (2020) (ref. 17), and, therefore that reference should appear at the end of the sentence.

Line 394-395: The highlighted sentence should be changed or deleted. According to the in-depth genetic study of human and dog parasites carried out by Thiele et al (2018) (reference 13), dog dracunculiasis does not represent any novel host switch but, otherwise, a historically large and stable population.  Therefore, the lack of reported cases in dogs before 2012 does not mean absence of transmission, but absence of GWD surveillance (no epidemiological study aimed to detect dracunculiasis in animals was conducted before 2012).

Author Response

We thank this Reviewer for their thoughtful assessment of our work, for their attention to detail, and for their suggested revisions, which have undoubtedly improved our manuscript. Please find our detailed responses to the comments provided below:

Line 30: the larval stage corresponds to the parasite (Dracunculus) not to the disease (dracunculiasis).

Thank you for this suggestion – we have revised the manuscript (Line 30) accordingly.

Line 33: to avoid confusion with the larvae inside the copepod, or in line 30 "L3 larvae" and in line 33 "L1 larvae” is specified, or “larvae” in line 33 should be replaced by "embryos".

The biology and lifecycle of D. medinensis is complex and we thank the Reviewer for highlighting that this might be unclear. We elected to revise the manuscript using the latter of the two suggestions and it now specifies that the female worm release “embryos (i.e., L1 larvae)” when wounds are exposed to a water source (Lines 33 & 34).

Line 38: The highlighted sentence should be changed. Certainly, there is no vaccine or effective treatment, but, of course, there have been and there are several prophylactic measures to prevent GW disease. As a matter of fact, thanks to those effective prophylactic measures adopted by the GW eradication campaign (explained by the authors in 3.2. section), GWD was eradicated in the majority of the countries affected by the disease in the 1960's.

Thank you for raising this point. We agree that it certainly is important to highlight that there are currently protective measures to prevent guinea worm infection – they are just not biomedical forms of prophylaxis (i.e., vaccines or medicines/drugs). We have revised the manuscript (Line 38) to make this point clear by specifying that “Neither a vaccine nor effective medical prophylaxis is currently available.”

Line 167: When listing the interventions implemented to reduce the burden of GWD in section 3.2., it would be more comprehensible if it were made in chronological order, i.e., with a clear distinction between the classical measures (those related to obtaining safe water) and those implemented very recently, due to, in turn, the recently accepted role of animals as reservoirs of the disease (those from Line 202: Tethering dogs and cats, the rewards, etc).

Thank you for this comment. Generally speaking, we do present the more ‘classical measures’ first followed by more recent interventions (e.g., tethering, cash rewards, anthelmintic treatments). However, while we can certainly appreciate the desire for cogency, we feel that, for our paper and its One Health lens, emphasizing how interventions can be cross-cutting and benefit both human and animal populations is more relevant to making those connections than presenting the interventions in precise chronological order.

Line 337: this assertion was already made by Galán-Puchades (2020) (ref. 17), and, therefore that reference should appear at the end of the sentence.

We appreciate the suggestion and have revised the manuscript accordingly (Line 339).

Line 394-395: The highlighted sentence should be changed or deleted. According to the in-depth genetic study of human and dog parasites carried out by Thiele et al (2018) (reference 13), dog dracunculiasis does not represent any novel host switch but, otherwise, a historically large and stable population.  Therefore, the lack of reported cases in dogs before 2012 does not mean absence of transmission, but absence of GWD surveillance (no epidemiological study aimed to detect dracunculiasis in animals was conducted before 2012).

We full-heartedly agree that the lack of reported cases in dogs prior to 2012 does not indicate the absence of transmission and was most likely due to insufficient surveillance systems. We have revised the manuscript in several locations, referencing Thiele et al.’s 2018 study where appropriate, to better make this point (Lines 142-144; Lines 398-400). We have also revised the lines in question from “epidemiological shift” to “apparent epidemiological shift” to emphasize that the shift may be ostensible (Line 397).

Reviewer 2 Report

Abstract: Delete this sentences which start  however to environmental health ( line 10,11,12,13).

Introduction. Delete lines (55, 56,57 and 58).

Results Delete fig 1

 Discussion. Please the discussion illustrate the results only but some time write comments far from the results of authors  

Author Response

We thank this Reviewer for their comments and suggested revisions. Please find detailed responses to the comments provided below:

Abstract: Delete this sentences which start however to environmental health (line 10,11,12,13).
We thank the Reviewer for this suggestion. We feel that the text contained in this section of the abstract is essential for the framing of the paper. It is indisputable that there is a growing body of evidence that animal populations are also affected by Dracunculus medinensis, which is clearly at odds with the final assumption stated. Further, if Dracunculus medinensis did exclusively infect humans, there would be little use for a One Health approach (defined/outlined in Lines 12 and 13). For these reasons, we have decided to leave these sentences in the manuscript.

Introduction. Delete lines (55, 56,57 and 58).
Thank you for this suggestion. However, outlined earlier and visualized in Figure 1, there is a growing body of evidence that while the number of human cases of GWD has decreased, there has been a substantial increase in the number of reported animal infections. Furthermore, there have been human and animal cases in new countries (e.g., Angola) and countries that had previously been certified free of GWD (e.g., Cameroon), which underscores the importance of surveillance and the ability to intervene in the case of infection in these areas. Accordingly, believe these lines to be of importance and have elected leave them in the manuscript.

Results Delete fig 1
We appreciate this suggestion but feel that Figure 1 is an important visual summary of the data reported in Table 1 and Table 2. For this reason, we have decided to leave the figure in the manuscript.

Discussion. Please the discussion illustrate the results only but some time write comments far from the results of authors
Thank you for this comment. We have used the Discussion section of our manuscript to discuss our results and frame them in the broader context of GWD control efforts, drawing on the work from other authors, where appropriate.

Reviewer 3 Report

The manuscript is focused on description of GWD, it zoonotic potential and different approaches for eradication among humans, animals and water as an important element of the ecosystem in the frame of One Health concept. Epidemiological data collected in the last five years are reported in six endemic countries, emphasizing the need for new research and interventions on animals and water, as well as the interaction between medical and veterinary services. The latest and comprehensive references have been used to elucidate the pathogenesis of the disease, the main hosts, transmission mechanisms, etc. On this basis are made conclusions and recommendations for a renewed approach for eradication of GWD in endemic countries including improved surveillance, dominant routs of transmission, risk factors and vectors among animals and humans, and last but not least new diagnostic and therapeutic approaches.  In this aspect I would recommend to be described and discussed the mostly applicable diagnostic methods and therapeutic strategies (with regards on their effectiveness) which are adequate for the endemic countries in this region of the world. Additionally, I would like to ask what is the role of wild animals from the families Canidae and Felidae in the GWD transmission and circulation in the environment? I agree with the overall statement concluding that despite the reduced number of cases in recent years and the ambition of the WHO, CDC, and others international organizations to eradicate this disease, this will not be possible in the next 10 years, given the many ambiguities surrounding the GVD.

Author Response

The manuscript is focused on description of GWD, it zoonotic potential and different approaches for eradication among humans, animals and water as an important element of the ecosystem in the frame of One Health concept. Epidemiological data collected in the last five years are reported in six endemic countries, emphasizing the need for new research and interventions on animals and water, as well as the interaction between medical and veterinary services. The latest and comprehensive references have been used to elucidate the pathogenesis of the disease, the main hosts, transmission mechanisms, etc. On this basis are made conclusions and recommendations for a renewed approach for eradication of GWD in endemic countries including improved surveillance, dominant routs of transmission, risk factors and vectors among animals and humans, and last but not least new diagnostic and therapeutic approaches.  In this aspect I would recommend to be described and discussed the mostly applicable diagnostic methods and therapeutic strategies (with regards on their effectiveness) which are adequate for the endemic countries in this region of the world. Additionally, I would like to ask what is the role of wild animals from the families Canidae and Felidae in the GWD transmission and circulation in the environment? I agree with the overall statement concluding that despite the reduced number of cases in recent years and the ambition of the WHO, CDC, and others international organizations to eradicate this disease, this will not be possible in the next 10 years, given the many ambiguities surrounding the GVD.

We thank the Reviewer for their evaluation of our manuscript and for their thoughtful suggestions. As outlined in the manuscript, currently, diagnosis in both human and animal populations relies on the visual detection of the worm at the skin surface. Further, at the present time, there are no effective medical therapeutics available for humans or animals. Anthelmintic treatments have been tested in dog populations, but did not demonstrate effectiveness in preventing or curing infections and were discontinued. With regard to the role of wild dogs and cats in GWD transmission and circulation in the environment, this remains unknown. Because domestic dogs have demonstrated competency as a host for Dracunculus medinensis, it is plausible that other members of the Canidae family (i.e., wild canids) may also play a role in transmission and environmental circulation. To the authors’ knowledge, this has not been studied. Cats have been speculated to play a lesser role in transmission because of their well-known aversion to immersion in water (Guinea Worm Wrap-Up #251). However, it is worth noting that not all cat species demonstrate the same aversion and guinea worms have been found in wild cats (i.e., a leopard, although this infection was not included in official case counts because it was detected while dissecting the animal and not at emergence, as is required for case counts). We have made revisions to the manuscript to emphasize this point (Lines 348 & 349) and feel that these intriguing questions underscore the importance of a One Health approach, which could help illuminate the answers to these queries.

Reviewer 4 Report

The manuscript by Dr. Boyce and colleagues provides a review of epidemiological evidence to advocate for a One Health approach in the fight against dracunculiasis, a neglected tropical disease that has been the target of eradication efforts since the 1980’s. Despite a marked decrease of reported cases in humans (from millions in the mid ‘80s to a few dozens in recent years), dracunculiasis has proved impervious to eradication, especially because D. medinensis, the parasite responsible for dracunculiasis, has recently been found to infect animal hosts in addition to humans. Starting from the available surveillance data and a review of the relevant literature, the authors discuss the interventions that are and/or could be used to reduce the burden of D. medinensis in humans and animals. The topic of this manuscript is interesting and fits well within the scope of the journal. The review seems well motivated and conducted, and the manuscript is clearly written and organized. For all these reasons, I only have relatively minor remarks, which the authors may want to address while revising their manuscript.

- I understand that this might be difficult to address, but there is no harm in asking: is there any way to quantify the surveillance effort in the countries/years shown in Table 1? The authors suggest that in some countries there have been no active searches for cases (p.2, l.56; p.3, l.103), but I wonder whether some quantification of surveillance effort may be available from others. This piece of information, albeit fragmentary or qualitative, would be of great importance for a proper commentary of the data shown in Table 1.

- A similar remark can be made about the increasing number of reports concerning infections in animals, most notably dogs. What is the best available evidence about that? Is it an actual trend, or is it something caused e.g. by increased surveillance in non-human hosts? In the former case, what are the possible explanations? (An increase in the size of the animal host population? Changes in human-animal interactions? Selective pressure for host-switching? Others?) Since infections in animals might be one of the most important factors that undermine eradication efforts, I think that this point deserves a more in-depth discussion within the present review.

- Among the factors that hindered the eradication of dracunculiasis, insufficient resources is not prominently analyzed here. I believe that some discussion of the economic aspects of the One Health approach could strengthen the overall message of this review. For instance, are there any cost-effectiveness studies about the eradication of dracunculiasis? Are there any economic tools, in addition to monetary incentives, that might assist in the last mile to the eradication of this parasite? Are there ways to promote the economic self-sufficiency of local actions, so as to favor the economic sustainability of the challenge? 

Minor comments

**************

- P.2, l. 80: “a list” is a bit vague, and could suggest that other countries might satisfy the inclusion criterion. I’d state more explicitly that these are the only countries that reported cases since 2015, if that is the case

Author Response

We thank this Reviewer for their thoughtful assessment of our work and for their valuable comments and suggestions revisions. Please find our detailed responses to the points raised below:

- I understand that this might be difficult to address, but there is no harm in asking: is there any way to quantify the surveillance effort in the countries/years shown in Table 1? The authors suggest that in some countries there have been no active searches for cases (p.2, l.56; p.3, l.103), but I wonder whether some quantification of surveillance effort may be available from others. This piece of information, albeit fragmentary or qualitative, would be of great importance for a proper commentary of the data shown in Table 1.

This is an interesting implementation question. Of the countries listed in Table 1, 5 have established eradication programs that are responsible for conducting surveillance activities (i.e., the Angolan Guinea Worm Eradication Programme, Chad's Guinea-Worm Eradication Programme, the Ethiopian Dracunculiasis Eradication Program, the Mali Guinea Worm Eradication Program, and the South Sudan Guinea Worm Eradication Program). Work has emphasized the need for ongoing active surveillance in these countries, as well as neighboring countries (especially Cameroon and Central African Republic), but to the authors’ knowledge, there has not been any work conducted to quantify their respective surveillance efforts. We agree that this represents an important piece of information that could hold important implications for GWD eradication efforts.

- A similar remark can be made about the increasing number of reports concerning infections in animals, most notably dogs. What is the best available evidence about that? Is it an actual trend, or is it something caused e.g. by increased surveillance in non-human hosts? In the former case, what are the possible explanations? (An increase in the size of the animal host population? Changes in human-animal interactions? Selective pressure for host-switching? Others?) Since infections in animals might be one of the most important factors that undermine eradication efforts, I think that this point deserves a more in-depth discussion within the present review.

We appreciate this perceptive comment. In 2018, Thiele and colleagues conducted a genetic analysis of Chadian Guinea worms that revealed that human and animal hosts share common parasite populations (reference 13). They concluded that while the observation of parasites emerging from animals may appear sudden, it is not novel within the history of the parasite and is indicative of historically stable transmission. All that to say, the lack of animal cases prior to 2012, especially in dogs, does not indicate the absence of transmission and was most likely due to insufficient surveillance. We have revised the manuscript in several locations, referencing Thiele et al.’s 2018 study where appropriate, to emphasize this point (Lines 142-144; Lines 398-400). We have also revised the text in our conclusion to emphasize that the observed epidemiological shift may be ostensible (Line 397).

- Among the factors that hindered the eradication of dracunculiasis, insufficient resources is not prominently analyzed here. I believe that some discussion of the economic aspects of the One Health approach could strengthen the overall message of this review. For instance, are there any cost-effectiveness studies about the eradication of dracunculiasis? Are there any economic tools, in addition to monetary incentives, that might assist in the last mile to the eradication of this parasite? Are there ways to promote the economic self-sufficiency of local actions, so as to favor the economic sustainability of the challenge? 

There has been little work done investigating the cost-effectiveness of Guinea worm eradication efforts. Fitzpatrick and colleagues published a 2017 study in PLoS Neglected Tropical Diseases, on the cost-effectiveness of Guinea worm eradication programs, but focused exclusively on human populations and on the aggregate effects of programs and not specific interventions. We emphatically agree that economic considerations regarding the GWD eradication efforts are important – especially if the concerning trend of animal infections continues, which would warrant new strategies and approaches – but feel that presently, there is insufficient evidence to discuss in this paper. We have revised the text to include economic analyses as topic worthy of future analyses (Lines 316-318).

- P.2, l. 80: “a list” is a bit vague, and could suggest that other countries might satisfy the inclusion criterion. I’d state more explicitly that these are the only countries that reported cases since 2015, if that is the case

In this instance, the “list [of sub-Saharan Africa countries that have reported GWD cases since 2015]” is synonymous with “catalog,” meaning that these are the only countries that have reported cases since 2015. We have revised the manuscript to try to make this more explicit (Line 80).